# Multimodality Treatment including Surgery Related to the Type of N2 Involvement in Locally Advanced Non-Small Cell Lung Cancer

**DOI:** 10.3390/cancers14071656

**Published:** 2022-03-25

**Authors:** Toon Allaeys, Lawek Berzenji, Patrick Lauwers, Suresh Krishan Yogeswaran, Jeroen M. H. Hendriks, Charlotte Billiet, Charlotte De Bondt, Paul E. Van Schil

**Affiliations:** 1Department of Thoracic and Vascular Surgery, Antwerp University Hospital, Drie Eikenstraat 655, 2650 Edegem, Belgium; toon.allaeys@student.uantwerpen.be (T.A.); lawek.berzenji@uza.be (L.B.); patrick.lauwers@uza.be (P.L.); sureshkrishan.yogeswaran@uza.be (S.K.Y.); jeroen.hendriks@uza.be (J.M.H.H.); 2Department of Radiation Oncology, Iridium Network, Faculty of Medicine and Health Sciences, University of Antwerp, 2610 Wilrijk, Belgium; charlotte.billiet@gza.be; 3Department of Pulmonology and Thoracic Oncology, Antwerp University Hospital, Drie Eikenstraat 655, 2650 Edegem, Belgium; charlotte.debondt@uza.be

**Keywords:** non-small cell lung cancer, chemotherapy, radiotherapy, chemoradiotherapy, concomitant, locally advanced, NSCLC, multimodality, immunotherapy, targeted therapy, surgery

## Abstract

**Simple Summary:**

Multimodality therapy for locally advanced non-small cell lung cancer (NSCLC) is a complex and controversial issue, especially regarding optimal treatment regimens for patients with ipsilateral positive mediastinal nodes (N2 disease). Is the landscape in this hotly debated stage changing the role for surgery as immunotherapy and targeted therapies are being investigated and implemented? A review on multimodality therapeutic options for stage IIIA-N2 NSCLC is presented.

**Abstract:**

For patients with locally advanced non-small cell lung cancer (NSCLC) or positive N1 nodes, multimodality treatment is indicated. However, the optimal management of patients presenting with ipsilateral positive mediastinal nodes (N2 disease) has not been determined yet. Different treatment regimens consisting of chemotherapy, radiation therapy, and surgery have been proposed and implemented previously. In more recent years, immunotherapy and targeted therapies have been added as therapeutic options. The role of surgery is currently redefined. Recent studies have shown that surgical resection after induction immunotherapy or targeted therapy is feasible and yields good short-term results. In this review, we summarize the latest data on multimodality treatment options for stage IIIA-N2 locally advanced NSCLC, depending on the extent of nodal involvement.

## 1. Introduction

With 1.8 million deaths in 2020 globally, lung cancer still remains the leading cause of cancer-related death [1]. Non-small cell lung cancer (NSCLC) accounts for the majority of lung cancer cases, up to 85%. At the time of diagnosis, most patients already present in an advanced stage [2]. In the locally advanced disease group, heterogeneity exists resulting in different prognostic subgroups for which different treatment strategies are proposed. An effort to solve this problem was provided by Goldstraw et al. in the latest, 8th edition of the Tumour, Node, and Metastasis (TNM) staging system of the American Joint Committee on Cancer (AJCC) where locally advanced stage III disease was further subdivided into stages IIIA, IIIB, and IIIC [3]. Loccoco et al. proposed to further divide this stage into resectable and unresectable disease [4]. The second ESMO (European Society for Medical Oncology) consensus conference on locally advanced NSCLC published in 2015 made a distinction between potentially resectable N2 and unresectable N2 disease [5]. In the recently updated ESMO clinical practice Guideline of 2021, stage IIIA has now been added under the group ‘early stages’. In stage III, a distinction is still made between resectable and unresectable disease [6]. Clearly defined definitions on technical resectability however, are lacking.

Despite these efforts, the ideal treatment regimens for stage III have not yet been defined, also due to the recent introduction of targeted therapies and immunotherapy as induction or adjuvant treatment. In contrast to the early and metastatic stages, no clear guidelines-based treatment algorithms are currently available.

Multidisciplinary teams in high volume centres are necessary to discuss the optimal treatment strategy for every individual presenting with locally advanced NSCLC. The role of surgery remains a controversial subject that has not been fully clarified [5]. Jeremic et al. were unable to find treatment-related predictive and prognostic factors in the management of stage III disease, more specifically in identifying patients eligible for surgery. This was mainly due to heterogeneity in factors reported in different studies [7]. Concurrent chemoradiotherapy (CRT) has traditionally been suggested as the gold standard, especially in the unresectable stages. Several multimodality regimens are currently recommended in patients with a good performance status, mainly consisting of induction chemotherapy, induction CRT and surgery, or definitive concurrent CRT [5]. In more recent years, many trials investigating immunotherapy and targeted therapies have changed the therapeutic landscape. In a recent update, the ESMO Clinical Practice Guidelines now even recommend adjuvant osimertinib for resected IB-IIIA NSCLC with epidermal growth factor receptor (EGFR) mutations, while in 2017, there was still no clear indication for targeted therapy in this non-metastatic setting [6]. The National Comprehensive Cancer Network (NCCN) also include this recommendation in their 2021 update [8]. Although promising trials on induction and adjuvant immune checkpoint inhibitors are highlighted in this update, no definite recommendations can yet be given [6]. In a previous review we performed on the feasibility of surgery after neoadjuvant immunotherapy and targeted therapy, promising results were found which will give rise to new treatment algorithms [9]. Implementation of these agents in daily practice will only be a matter of time. In this review, we summarize the surgical and non-surgical treatment strategies in locally advanced stage IIIA-N2 NSCLC.

## 2. Materials and Methods

A narrative literature review was conducted. A search on database MEDLINE and PubMed was carried out for literature research screening. References were identified using the following terms: non-small cell lung cancer, surgery, immunotherapy, radiotherapy, chemotherapy, targeted therapy and locally advanced OR stage III and N2 disease OR mediastinal involvement for the interval of 2005 till November 2021. Only articles published in the English language were selected. The highest level of evidence was sought for selecting only randomized controlled trials, systematic reviews, narrative reviews and meta-analysis in our search. Other relevant studies or articles referred to in these articles were critically screened and included when considered suitable for contribution. Endnote^®^ software was used to cite the references in this manuscript.

## 3. Stage IIIA-N2 Disease

This subgroup of locally advanced disease is still hotly debated. Various multimodality treatments are described, where the role of surgery is often being questioned. No predictors nor prognosticators were identified in a review on the management of stage IIIA-N2 disease, not facilitating the decision-making progress regarding the role of surgery in this subgroup [7]. Another difficult point remains the distinction between resectable and unresectable disease as no clear definitions exist. Furthermore, nodal status represents a reliable and important prognostic predictor in our daily clinical practice. However, mediastinal nodal involvement is not a single entity and comprises different subcategories ranging from unexpected N2 involvement to intranodal resectable disease and bulky extranodal involvement with encasement of the large vessels. Decaluwé et al. highlighted this issue in a retrospective study of 92 patients with pathologically proven N2 disease. Patients were included if a response or stable disease was present after induction chemotherapy. A better survival was observed in patients with mediastinal nodal downstaging compared to those with persistent N2 disease, although not significantly. When considering single N2 or multiple N2 persistent disease at the time of resection, a significant difference in 5-year survival (5YS) was observed in favour of the former [10]. This is consistent with data from literature [11,12,13] such as the IASLC staging project showing a 5YS of 34% and 20%, respectively [11]. Furthermore, no significant difference was found between patients with persistent, single N2 entity and patients with mediastinal downstaging to N1 [10]. Asamura et al. for the International Association for the Study of Lung Cancer (IASLC) even went further and proposed several subcategories regarding the nodal status in TNM staging [14]. Proposals such as counting the number of metastatic lymph nodes and further dividing the N2 nodal status into N2a and N2b were made. N2a representing single station involvement is further split into N2a1, where there is no metastatic hilar or pulmonary N1 positive lymph node, and N2a2, when N1 involvement is also present. The former is the so-called skip N2 disease, having a better prognosis than N1b (multiple N1 stations involved), although not significantly so. On the other hand, survival of N1b (multiple N1 nodes involved) is similar to N2a2 [14]. Skip N2 disease is a better predictor of disease-free survival (DFS), together with other descriptors such as number of involved lymph nodes, and lymph node ratio (LNR) which is defined as the number of positive nodes/total number of resected nodes [13].

In patients eligible for surgery, the role of complete resection is still of uttermost importance. As a subcommittee of the IASLC created in 2001, the Complete Resection Subcommittee proposed an internationally accepted definition on complete resection in lung surgery. This includes the following: free resection margins proved microscopically, systematic nodal dissection or lobe-specific systematic nodal dissection with at least 6 nodal stations (3 N1 and 3 N2 stations including subcarinal station no.7), no extracapsular nodal extension of those nodes removed separately or at the margin of the lung specimen, and the highest mediastinal lymph node should be negative [15].

### 3.1. Unforeseen (Unexpected or Surprise) N2

With preoperative staging techniques becoming more accurate, unexpected N2 disease is rare. A recent meta-analysis on the benefit of confirmatory mediastinoscopy after a negative endobronchial ultrasound (EBUS) in staging patients with resectable NSCLC showed a similar rate of involvement of 9.6% [16]. Adjuvant chemotherapy should be administered to reduce the risk of locoregional or distant relapse due to micrometastases. Postoperative sequential radiotherapy (PORT) is not indicated in case of all criteria of achieving a complete resection being accomplished [5]. This statement from the 2015 ESMO guidelines was just recently confirmed by the long-awaited LungART randomised trial [17]. In this trial, a total of 501 patients with N2 disease who underwent complete resection were randomized between PORT or not. They mostly received neoadjuvant or adjuvant chemotherapy. Results showed a non-significant difference in DFS of 15% in favour of patients receiving PORT. Currently, PORT is not advised in the adjuvant setting for completely resected N2 disease [17]. In patients with stage II and III disease and positive resection margins, however, PORT offers a benefit in OS, irrespective of nodal stage. [18].

Based on this evidence, we provide the current treatment algorithm in our institution for unforeseen N2 disease (Figure 1).

### 3.2. Resectable N2

#### 3.2.1. Induction Chemo(radio)therapy

Definition of resectability in N2 disease is still vague. Multidisciplinary teams, including an experienced thoracic surgeon as a member, are necessary to evaluate every individual case.

The 2017 ESMO guidelines consider single station N2 disease as resectable. In that case, neoadjuvant chemotherapy is advised. If mediastinal downstaging is obtained after induction treatment, surgery is advised if pneumonectomy can be avoided. Treatment options include surgery after neoadjuvant chemotherapy or chemoradiotherapy [19].

The intergroup trial 0139 is a phase III study randomizing patients with N2 disease between either neoadjuvant chemoradiotherapy plus surgery or either chemoradiotherapy alone. A total of 164 patients underwent surgery. Although not statistically significant, patients in the surgery arm were reported to have a better progression-free survival (PFS). Overall survival (OS) was not different between both arms, mainly due to the lack of power and a reduced number of chemotherapy cycles that could be given in the surgical arm, but also because of the high mortality rate for patients undergoing a pneumonectomy. Therefore, an exploratory analysis was conducted showing improved OS in patients undergoing lobectomy compared to a matched group treated by chemoradiotherapy alone. The study concluded both therapies can be suggested in this subgroup of patients, especially when lobectomy is likely. Interesting to mention is that in both arms, up to 75% were reported to have a pretreatment single level N2 disease [20].

Another interesting phase III trial is the EORTC 08941 study, targeting initially unresectable stage IIIA-N2 locally advanced NSCLC. Induction chemotherapy was given, followed by either surgery or radiotherapy if at least a minor response was present. In the surgery arm, 167 patients were included. No difference regarding OS and DFS was observed between the two arms. A statistical difference, however, in 5YS was found between patients undergoing a lobectomy versus pneumonectomy in favour of the former, the same trend being found in the Intergroup trial. The authors suggest radiotherapy rather than surgery in case of response to induction chemotherapy in this group of patients. Complete resection was only achieved in 50% of patients using the strict criteria of the International Association for the Study of Lung Cancer (IASLC) [15]. Keeping the recently published LungART trial in mind, this high rate of R1 resections may have influenced the results of this study. Furthermore, the standard of concurrent chemoradiotherapy was not used as a control arm. Whether surgery is beneficial in patients having mediastinal clearance after induction chemotherapy or chemo-immunotherapy has yet to be investigated [21].

Regarding resectable N2 disease, patients with multiple stations of mediastinal nodal involvement are the most difficult subgroup of stage IIIA-N2 patients. In the previously mentioned Intergroup trial 0139, a better PFS was found in patients undergoing surgery, especially when a lobectomy was performed. Of all patients included, 22% were reported to have more than 2 nodal stations involved. [20] In the ESPATUE trial, 264 patients with resectable locally advanced NSCLC were included for induction chemotherapy and chemoradiotherapy. Finally, 161 were randomized to either surgery or boost concurrent chemoradiotherapy. In both arms, only one third of patients were considered to have potentially resectable N2 disease. No further details are available on the specific mediastinal nodal stations involved. Both strategies were deemed equivalent because no difference in 5YS nor PFS was reported [22].

The multicentered SAKK 16/00 trial by the Swiss Group for Clinical Cancer Research (SAKK) enrolled 232 patients with pathologically proven stage IIIA-N2 disease for surgery after being randomized to either neoadjuvant chemotherapy alone or neoadjuvant sequential chemoradiotherapy. Almost all patients were reported to have no more than 5 cm bulky mediastinal nodal involvement. Further details, however, on the number of stations involved are lacking. In the group of patients receiving additional neoadjuvant radiotherapy, a trend towards a better response to induction treatment with more nodal downstaging, and a higher rate of complete resection and pathological complete response (pCR) was observed, which was, however, not significant. In addition, no difference was found in event-free survival (EFS) and OS between both groups. So, radiotherapy did not add any benefit compared to neoadjuvant chemotherapy alone in resectable stage IIIA-N2 disease [23]. However, it should be noted that sequential chemoradiation was administered instead of concurrent chemoradiation, which is considered standard of care for stage III NSCLC.

When looking at meta-analyses of multimodality treatment regimens in stage IIIA-N2 disease, surgery still plays a role. Recently, Zhao et al. included 18 randomized controlled trials with 13 different treatment regimens. They concluded that neoadjuvant chemotherapy with surgery followed by either adjuvant chemotherapy or radiotherapy provides the best possible treatment option regarding overall survival and treatment-related deaths in patients with stage IIIA-N2 disease [24].

McElnay et al. included six trials in their meta-analysis, of which four included induction chemotherapy and two, induction chemoradiotherapy. In the trimodality setting, a relative improvement of 13% in OS was found in patients receiving surgery versus chemoradiotherapy alone. The authors concluded that surgery was preferred in a trimodality treatment regimen for resectable stage IIIA-N2 disease. This statement does not fully hold anymore, as an update in 2019 correctly states that the confidence interval of no effect was crossed. However, surgery can be considered in both the bimodality as trimodality regimens as survival outcomes were similar [25].

However, the 2019 NICE guidelines still recommend chemoradiotherapy plus surgery 3 to 5 weeks later in patients with resectable stage IIIA-N2 disease as it improves PFS and possibly OS. Furthermore, this trimodality treatment has shown to be more cost effective than chemoradiotherapy alone [26].

Based on the current evidence, we provide the current treatment algorithm in our institution for resectable N2 disease (Figure 2).

#### 3.2.2. Adjuvant Radiotherapy

When looking at the role of postoperative radiotherapy (PORT) in patients with resected stage IIIA disease, two large studies were recently published. Analysing the SEER database in which 1711 out of 5168 patients underwent PORT, this was shown to improve OS in patients with N2 disease having 6 or more lymph nodes involved. Details on percentage of R0 is, however, unknown [27].

Recently, the long-awaited LungART trial was published showing no higher disease-free survival rates compared with no PORT [17]. In patients with stage II and III disease and positive resection margins, however, PORT offers a benefit in OS, irrespective of nodal stage [18].

#### 3.2.3. Induction Immunotherapy

With the emerging immunotherapy and targeted therapies as novel agents, several interesting trials must be mentioned especially in the neoadjuvant setting (Table 1). Overall survival has been the gold standard for evaluating the clinical benefit in clinical trials on cancer treatment. Major pathological response (MPR), defined as less than 10% residual viable tumour cells in the resected specimen, had been accepted as surrogate endpoint for survival in neoadjuvant chemotherapy [28,29]. Both MPR and pCR have now been generally accepted as surrogate endpoints for survival in neoadjuvant immunotherapy for resectable NSCLC. However, it has not been validated and therefore, further studies are needed [28,29,30,31,32,33].

The LCMC3 trial investigated atezolizumab as monotherapy prior to surgery in a phase II study. In total, 181 patients with stage IB-IIIB were included, of which stage IIIA included 39% of cases. MPR and pCR were 20% and 7%, respectively. No concerns were reported regarding safety and surgical feasibility. MPR was positively associated with PD-L1 (programmed death-ligand 1) expression and negatively associated with EGFR/ALK mutations [34].

As these regimens are just recently being investigated in experimental studies, most trials do not make further distinction between early-stage disease and resectable stage IIIA disease, nor are different subgroups of mediastinal nodal involvement studied.

Some studies do investigate this specific subgroup of N2 disease in stage IIIA NSCLC, such as the SAKK16/14 trial by Rothschild et al., where patients with resectable stage IIIA-N2 disease were enrolled. Neoadjuvant chemotherapy was administered, followed by sequential durvalumab prior to surgery. Surgery was performed in 55 of the 68 enrolled patients. Postoperative downstaging to N0-N1 was observed in 67%. MPR and pCR were 62% and 18%, respectively. MPR and nodal downstaging were not influenced by pretreatment PD-L1 expression. The 1-year EFS rate was 73%. The authors concluded this combination to be superior to chemotherapy alone, given the results of 1-year EFS 48% in the latter group [35].

In the NADIM trial, a phase II, single arm, multi-centre study, 46 eligible patients were enrolled having resectable stage IIIA disease, including N2 involvement. Neoadjuvant immunotherapy using nivolumab and platinum-based chemotherapy were administered, followed by surgery and adjuvant nivolumab. Almost all patients dealt with treatment-related adverse events (tAEs), of which almost a third showed grade 3 toxicities or higher. In all patients undergoing surgery (41/46), complete resection was achieved. No delays or surgical deaths were reported. Response to induction treatment was found in three quarters of patients with zero cases of progressive disease. Pathological complete response (pCR) was observed in 83%. The two-year PFS and OS were 77.1% and 89.9%, respectively. Of patients with pCR, almost all were progression-free at two years follow-up [36].

The checkmate 816 trial, a randomized phase III trial, investigated neoadjuvant nivolumab in addition to chemotherapy compared to chemotherapy alone in 358 patients with resectable NSCLC without EGFR/ALK mutations. Almost two third of included patients had stage IIIA disease. This study showed a better pathological response with the combination of neoadjuvant nivolumab and chemotherapy. The primary endpoint of pCR was significantly improved from 2.2% to 24%, regardless of disease stage, histology, and PD-L1 expression levels. MPR were 8.9% and 36.9%, respectively. When looking at the bimodal arm, pCR was higher with PD-L1 levels of ≥1%. The second primary endpoint of EFS was also found to be positive in a recent press release (not published yet). Surgical outcomes were extensively reviewed and regarded acceptable. Duration of surgery, rates of R0, extent of resection, tAEs, complications, length of stay were not affected by the addition of nivolumab. This trial may change the landscape of resectable NSCLC as neoadjuvant immunochemotherapy could become standard of care [37].

Similar results with neoadjuvant chemoimmunotherapy regarding pCR, R0 resection rate and feasibility were seen in two smaller, prospective studies with a pCR of 45.9% and 29.1%, respectively, in 37 patients with stage IIB-IIIB disease, and 72 patients with stage IIIA disease [38,39].

In their phase II trial, Shu et al. enrolled 30 patients with stage IB-IIIA disease, of which 77% were in the latter subgroup. Patients were given neoadjuvant atezolizumab with chemotherapy. A MPR of 57% was achieved [40].

Another trial using nivolumab with chemotherapy as induction treatment for patients with stage IB-IIIA disease showed good pathological response with MPR in 11/30 (85%) patients, of which 5 were with pCR. The former was not influenced by PD-L1 levels [41].

Placebo versus pembrolizumab in addition to neoadjuvant chemotherapy is being investigated in the phase III KEYNOTE-671 trial for patients with resectable IIB-IIIA disease [42].

Placebo versus atezolizumab in addition to neoadjuvant chemotherapy is being investigated in the phase III Impower030 trial for patients with resectable II-IIIB disease [43]. Placebo versus nivolumab both in neoadjuvant setting with chemotherapy as in adjuvant setting is investigated in the checkmate 77T trial for patients with stages IIA-IIIB. EFS is the primary endpoint [44].

Consequent to the results of the NADIM trial, the NADIM II (NCT03838159) trial was set up. This phase II, multi-centre trial includes nivolumab in the neoadjuvant regimen together with chemotherapy as well as in the adjuvant setting for 6 months compared to neoadjuvant chemotherapy alone. This study looks for pCR in patients with resectable locally advanced NSCLC.

In the AEGEAN trial, another phase III trial, the efficacy of durvalumab versus placebo added as well to neoadjuvant chemotherapy as in the adjuvant setting for resectable stage IIA-IIIB NSCLC is investigated in an estimated 300 patients. MPR is the primary endpoint [45].

Combination of immune-checkpoint inhibitors (ICI) is studied as well. The NEOSTAR study, a phase II trial, investigated nivolumab vs. nivolumab plus ipilimumab in a neoadjuvant setting in 44 patients with resectable stage I-IIIA disease. Major pathological response was the primary endpoint, which was 24% and 50%, respectively. The combination of ICI also resulted in a higher pCR of 38% versus 10% in the nivolumab arm [46].

Many trials are ongoing looking for the benefit of combining neoadjuvant immunotherapy with radiotherapy for resectable NSCLC. Evidence is emerging establishing a rationale for the future combining of these therapies. Deng et al. showed in its mice model the upregulation of PD-L1 after radiotherapy. Furthermore, the therapeutic effect of radiotherapy was enhanced when administrating anti-PD-L1 therapy [47]. Many other studies in other solid tumours have published identical studies favouring the combination of immunotherapy with radiation [48]. As neo-adjuvant immunotherapy is promising but still warrants further research for identifying the optimal treatment schemes, adding radiotherapy seems to be even more promising but makes this issue even more complex. Patient selection, the ideal treatment regimen such as timing, dose, and location, still needs further research. When considering feasibility and safety of radiotherapy, there are no additional toxicities when combined with immunotherapy [49]. One phase II trial recently published preliminary results in 60 patients with stage I-IIIA NSCLC disease. MPR was significantly improved in the combination arm, being 53.3% compared to 6.7% in the durvalumab only arm. Of the 16/30 patients with MPR in the combination group, 50% had pCR [50].

Two ongoing trials are assessing neoadjuvant immunotherapy with CRT, followed by surgery and adjuvant immunotherapy in patients with stage IIIA-N2 disease. The Hoosier Cancer Research Network (NCT03871153) explores durvalumab in 25 patients with pCR as primary endpoint. The Case Comprehensive Cancer Center (NCT02987998) is evaluating pembrolizumab in 9 patients in a phase I trial, assessing safety as primary endpoint.

Although promising results are achieved with neoadjuvant immunotherapy, two studies reported high morbidity and mortality rates. Bott et al. encountered a high conversion rate of 54% because of hilar inflammation and fibrosis following neoadjuvant nivolumab in a phase I trial [51]. The IFCT-1601 IONESCO trial evaluating induction durvalumab in patients with IB-IIIA non-N2 disease was stopped early because of a high 90-day mortality rate of 9%. A pneumonectomy was necessary in 20.5% [52].

However, neoadjuvant immunotherapy was deemed feasible by the authors in a previous review [9].

#### 3.2.4. Adjuvant Immunotherapy

Several trials are investigating immunotherapy in an adjuvant setting in resectable NSCLC.

The IMpower010 is a randomized, multicenter, phase III study investigating the benefit of adjuvant atezolizumab compared to best supportive care in patients with resectable stage IB-IIIA disease following surgery and adjuvant chemotherapy. A total of 1280 patients were enrolled of which 1005 were eligible for randomization. A benefit in DFS was seen in favour of adjuvant atezolizumab, especially in the subgroup of patients with PD-L1 expression levels of more than 1% where a 34% reduction in risk of disease recurrence or death is reported. This benefit is the highest in patients with levels of more than 50%. In most patients with significant expression of PD-L1 levels, the absence of EGFR/ALK mutations resulted in a higher DFS hazard ratio. Minimal differences regarding relapse are found between the two study arms; however, in patients with PD-L1 levels ≥1%, time to relapse appeared to be longer in the atezolizumab arm. These results are promising, though longer follow up is warranted to obtain valid overall survival (OS) data [53].

Three other phase III trials are investigating the role of adjuvant immunotherapy and its effect on DFS [54]. In the ANVIL trial, adjuvant nivolumab in resected stage IB-IIIA disease is compared to standard of care observation [55].

Adjuvant pembrolizumab following surgery and adjuvant chemotherapy is compared to placebo for patients with stage IB-IIIA disease in the PEARLS/Keynote-091 trial. DFS is prospectively investigated in these patients, especially looking for differences in subgroups with different PD-L1 expression levels [56].

Durvalumab versus placebo following complete resection in stage IB-IIIA disease, including 1360 patients, is another interesting phase III trial (NCT 02273375). DFS is the primary endpoint.

In a phase II, single-arm trial, consolidation with pembrolizumab was investigated in 37 patients with N2 disease after concurrent chemoradiation and surgery. DFS of more than 20 months was not reached, though the full results of the trial are yet to be released [57].

#### 3.2.5. Targeted Therapies

Several studies on the use of the tyrosine kinase inhibitor (TKI) in a neoadjuvant setting in patients with resectable stage IIIA-N2 disease show promising results.

In the EMERGING CTONG 1103 trial, 71 patients with stage IIIA-N2 disease with an EGFR-mutation were enrolled and randomized to either erlotinib or cisplatin plus gemcitabine chemotherapy as induction therapy.

First of all, erlotinib seemed feasible, as no major tAEs were reported compared to an almost one third prevalence in the chemotherapy group. The primary endpoint of objective response rate was not met (54.1% and 34.3%, respectively). However, a significantly better PFS of 21.5 months versus 11.4 months was found in the erlotinib group, but in a recent update, no difference in OS was observed between both groups [58].

Another trial, however, was not able to show any survival benefit in patients receiving erlotinib compared to chemotherapy, but an improved response rate to induction treatment was seen in favour of the former. However, in this phase 2 study, only 24 patients were included [59].

In a third, phase II trial on neoadjuvant erlotinib, an improved radical resection rate was noted in 14 out of 19 patients receiving surgery. Performing next generation sequencing in a non-metastatic setting seems increasingly valuable as this may possibly change the use of preoperative TKIs in the future, as the presence of an additional high burden of TP53 mutation in their cohort was associated with a worse PFS (8 months) compared to patients with no, or very low prevalence of TP53 mutation (36 and 38 months, respectively) [60].

A meta-analysis on preoperative TKI’s by Sun et al. concluded their use to be feasible. In a subgroup analysis of 68 patients with N2 disease extracted from three studies, mediastinal downgrading and pCR were 14% and 0%, respectively [61]. For comparison, in the EMERGING CTONG 1103 trial, no pCR was seen in either arm [58].

In the ADAURA trial, 682 patients with stage IB-IIIA and EGFR mutations were enrolled in a phase III, double-blind randomized controlled trial. Adjuvant osimertinib versus placebo was administered in patients after complete resection followed by adjuvant chemotherapy. In stage II-IIIA patients, a significant difference in two-year DFS was seen favouring osimertinib versus placebo (90% vs. 44%) [62]. Adjuvant Osimertinib is now recommended in both the 2021 updates of the ESMO Clinical Practice Guidelines as well as the National Comprehensive Cancer Network (NCCN) for this subgroup of patients [6,8].

Some trials are still ongoing. In the NeoADAURA trial, patients with resectable II-IIIB N2 disease will be enrolled for either neoadjuvant osimertinib with or without chemotherapy versus neoadjuvant chemotherapy. MPR is the primary endpoint [63].

In the NCT03433469, neoadjuvant osimertinib in resectable stage I-IIIA disease as single arm is investigated using MPR as primary endpoint. NCT04302025 will be enrolling 60 patients for which different targeted therapies are scheduled both as neoadjuvant as adjuvant therapy, in addition to adjuvant chemotherapy.

### 3.3. Unresectable N2

Locally advanced disease is considered unresectable if no complete resection (R0) can be achieved, even after induction therapy [19]. Mediastinal lymph nodes that are bulky, cannot be individualised, or those encasing major structures are considered as unresectable N2 disease.

The treatment of choice in unresectable stage IIIA-IIIB is concurrent chemoradiation. If not possible, sequential chemoradiation is advised. Two to four cycles of cisplatin-based chemotherapy with 60–66 Gy in 30–33 daily fractions are the recommended scheme [19]. In the recent 2021 update of both ESMO and NCCN, durvalumab as consolidation therapy is advised in patients with PD-L1 levels ≥ 1% if no progression is noted [6,8]. This recommendation was based on the phase III PACIFIC trial randomizing 713 patients to placebo or durvalumab for one year within the first 42 days after concurrent chemoradiotherapy, if no progressive disease was present [61]. Both PFS and OS were significantly higher in the durvalumab arm. Notably, PFS was higher when the drug was administered within two weeks after radiation therapy compared to when there was a delay of more than two weeks. Warranting further research is the benefit of durvalumab in patients with EGFR mutations as no benefit of durvalumab was noted in those patients with PD-L1 levels lower than 25% [64].

Similar results regarding PFS and OS were seen using pembrolizumab as consolidation therapy in the LUN 14-179 trial, a phase II study. Morbidity was similar in those treated with chemoradiation alone [65].

A phase I trial investigated pembrolizumab in combination with neoadjuvant concurrent CRT in 21 patients and concluded it to be feasible and promising as a median PFS of 69.7% at one year was reached for patients receiving at least one dose [66].

The NICOLAS trial, a phase II, single arm trial investigated the concomitant use of nivolumab with concurrent CRT as neoadjuvant treatment in patients with stage III NSCLC. Just recently, results were published regarding its efficacy. In this trial, with almost two third of patients having stage IIIB, a one-year PFS of 53.7% was reported. The two-year OS in stage IIIA patients was 81% [67]. This regimen was considered feasible and safe in an earlier report by the authors [68].

In the AFT-16 trial, neoadjuvant and adjuvant atezolizumab in addition to concurrent CRT showed promising results regarding safety and efficacy. The one-year PFS was 66% but further endpoints are yet to be further analysed [69].

Another ICI, atezolizumab, is being investigated in the phase II DETERRED trial where it is either used as consolidation drug only following CRT or both concurrently with CRT and in the consolidation phase. The concurrent administration is deemed safe. Results on efficacy are still awaited [70].

As durvalumab as consolidation in the PACIFIC trial resulted in a new recommendation, and the addition of ICI to standard of care seems promising, the possible additional benefit of durvalumab in the concurrent setting with the standard of care for unresectable locally advanced NSCLC is now being investigated in two large phase III, double-armed, multicentre trials. In the PACIFIC2 trial, durvalumab in a concurrent setting with CRT is compared with CRT plus placebo [71]. In the EA5181 trial, durvalumab in a concurrent setting with CRT is compared with consolidation durvalumab only [72].

Besides these trials, many studies are investigating possible novel regimens in this subgroup. Such a study is the ongoing checkmate 73L trial. This multicentre, phase III trial will include a total of 888 patients having unresectable, locally advanced NSCLC. Patients will be randomized 1:1:1 resulting in an arm with nivolumab and CRT in the neoadjuvant setting, followed by nivolumab and ipilimumab (arm A) or nivolumab alone (arm B) and an arm with neoadjuvant CRT followed by durvalumab. Endpoints are PFS and OS [73].

Many other promising trials are ongoing for this subgroup of patients, which will aid in elucidating the role of ICI in unresectable locally advanced NSCLC (Table 2). A recent review by Käsmann et al., listing all ongoing trials covering this hot topic, concluded that ICI will play a fundamental role for these patients in the future though many questions such as the duration of consolidation therapy are yet to be answered [74].

Based on the current evidence, we provide the current treatment algorithm in our institution for unresectable N2 disease (Figure 3).

## 4. Discussion

Clear, uniform guidelines on the management of stage IIIA-N2 disease are lacking. One of the main reasons is the heterogeneity of this subgroup. Mediastinal nodal involvement can range from unforeseen N2 involvement to skip N2 and unresectable N2 disease. These presentations are correlated with a different prognosis and prompt their individual management. Multidisciplinary teams are of paramount importance in discussing each individual case. Technical resectability, feasibility of R0 resection, resection limited to lobectomy are all factors to be considered by at least one experienced thoracic surgeon. While some authors are further subdividing N2 disease, some are still not providing details on this important feature in their clinical trials. Asamura et al. went further and proposed several subcategories regarding the mediastinal nodal status in TNM staging such as N2a1 and N2a2 based on presence of N1 involvement in the case of a single N2 station [14]. Generally accepted is the subdivision of N2 involvement into unforeseen N2 disease, single station N2 disease, and multiple station N2 disease. The latter is subdivided into potentially resectable and unresectable N2 involvement. As mentioned earlier, definitions on resectability are lacking. Intranodal, single station N2 is generally accepted as resectable whereas multiple bulky N2 nodes, especially when encasing major structures, are unresectable. According to NCCN, multiple proven N2 stations larger than 3 cm are considered unresectable [75]. In between is the grey zone, which is a subject for discussion. For single station N2 and potentially resectable multiple station N2 disease, multimodality treatment including surgery is advised. Studies like the Intergroup trial 0139 trial and EORTC 08941 did not find any statistical benefit favouring surgery compared to CRT alone, and sequential radiotherapy following neoadjuvant chemotherapy, respectively. However, in the latter, only patients with initially unresectable NSCLC were included. Furthermore, when performing an exploratory analysis, a better trend towards survival was observed in patients receiving a lobectomy versus pneumonectomy. Caution is advised regarding recommendations, but it seems that, if a lobectomy can be achieved, surgery might be a better option. The 2019 NICE guidelines recommend trimodality treatment including chemoradiotherapy plus surgery in patients with resectable stage IIIA-N2 disease. Improved PFS and a not statistically improved OS are observed in this group of patients. Cost-effectiveness also seemed to favour this trimodal approach [26]. However, as proven in the SAKK trial, no additional benefit was seen when induction radiotherapy was added to the scheme of neoadjuvant chemotherapy for resectable stage IIIA disease, though administrated sequentially instead of concurrently [23]. As uniform guidelines advising neoadjuvant chemoradiation are lacking, we are cautious and propose induction chemotherapy as standard in our institution. In the third version of the 2020 guidelines of the NCCN, induction treatment by either chemotherapy or CRT is advised for resectable stage IIIA-N2 disease. When response to induction treatment is observed, surgery is to be considered [75]. Concurrent CRT is a valid alternative according to this guideline. An update in 2019 of the meta-analysis of McElnay et al. no longer showed a better survival rate in patients with trimodality regimen compared to CRT alone. However, surgery can be considered in both the bimodality as trimodality regimens as survival outcomes were similar [25]. The British Thoracic Society Lung Cancer Specialist Advisory group advises both trimodality and bimodality treatment regimens in patients with resectable stage IIIA-N2 disease, irrespective of the number of N2 nodal stations involved. Bimodality therapy included either radiotherapy or surgery after induction chemotherapy [76].

In Switzerland, daily practice treatment regimens are based on the extent of mediastinal nodal involvement. In case of single node involvement, surgery is preferred as part of a multimodality treatment. Radiotherapy is preferred over surgery in case of bulky nodes and/or increasing number of N2 nodal stations involved [77].

The role of postoperative radiotherapy in stage III-N2 disease has always been controversial. However, the recently published LungArt trial has clearly demonstrated that PORT has no additional benefit in completely resected N2 disease [17]. In case of positive resection margins, however, PORT still offers a benefit in OS [18]. In an analysis of the SEER database, the same result was observed in N2 disease with 6 or more lymph nodes involved and treated by PORT; however, details on R0 resection are lacking [27].

The new era of immunotherapy and targeted therapies has not yet resulted in more definite recommendations for stage IIIA-N2 disease.

Authors evaluating the safety of neoadjuvant immunotherapy in locally advanced NSCLC conclude that it is safe and feasible [9]. When assessing its efficacy, many mention the need for introduction and development of biomarkers. Overall survival still remains the gold standard for evaluating new cancer treatments in clinical trials, but as this takes years to accomplish, major pathological response was developed as the surrogate endpoint for neoadjuvant chemotherapy [28,29]. Both MPR and pCR have now been generally accepted as surrogate endpoints for survival in neoadjuvant immunotherapy for resectable NSCLC. However, they have not yet been validated and therefore, further studies are needed [28,29,30,31,32,33].

Both MPR and pCR showed excellent results following neoadjuvant immunotherapy. Many trials are still ongoing. Their results will probably change the multimodality landscape in stage IIIA-N2 disease. Neoadjuvant immunochemotherapy is considered the most established and promising regimen [31,33]. The NADIM trial, offering neoadjuvant nivolumab in addition to platinum-based chemotherapy for patients with stage IIIA-N2 disease, indeed showed excellent results with a pCR up to 83% [36]. Many ongoing trials such as the NADIM II trial, KEYNOTE-671, 77T trial, IMPower030 and AEGEAN trial, comparing induction immunochemotherapy with chemotherapy alone, are promising. The investigators of the SAKK 16/14 trial have already published their results with this setup including durvalumab in patients with stage IIIA-N2 disease [35]. Postoperative downstaging to N0–N1 was observed in 67%. MPR and pCR were 62% and 18%, respectively. The 1-year EFS rate was 73%, compared to 48% in the chemotherapy arm [35].

Another example showing the superiority of induction chemoimmunotherapy compared to chemotherapy alone is the Checkmate 816 trial showing an improved pCR from 2.2% to 24%, regardless of disease stage, histology, and PD-L1 expression levels. MPR was 8.9% and 36.9%, respectively. In a press release, EFS also favours the combination therapy. Surgical outcomes such as duration of surgery, rates of R0, extent of resection, tAEs, complications, length of stay were deemed acceptable and not inferior when adding nivolumab [37].

Whether induction with multiple ICIs is better is still being investigated (NCT02998528) [33]. The combination of multiple ICI was set up in the NEOSTAR study, with nivolumab plus ipilimumab being superior to nivolumab alone, resulting in a higher MPR (50% vs. 24%) and pCR (38% vs. 10%) [46]. Acceptable toxicity rates were reported in this trial whereas a recent phase I study by Reuss et al. was terminated earlier because of high toxicity rates. Of the nine enrolled patients, six patients (67%) showed tAEs, of which fifty percent was grade 3 or higher. Multiple reasons were proposed, such as small sample size, duration between administration and surgery, and a high rate of mutations [78]. Future studies, however, are needed to further elucidate the role of combining ICIs and its safety.

Adding radiotherapy to immunotherapy seems promising, as PD-L1 upregulation was seen in mice due to irradiation [47]. Combining these two therapies seems to provide a large benefit with acceptable toxicities [47,48,49]. The phase II trial in NSCLC showed some promising preliminary results of adding radiotherapy to induction durvalumab increasing MPR significantly from 6.7% to 53.3% compared to durvalumab alone [50]. Further results on this regimen, however, are pending. In general, this combination makes the ideal treatment scheme for patients with locally advanced lung cancer more promising, though also more complex to fine-tune.

In the adjuvant setting, immunotherapy seems to be promising as well. The IMPower010 trial showed a better DFS with adjuvant immunotherapy compared to chemotherapy in resected stage IB-IIIA patients [55]. Many other trials are pending, some specific for N2 disease [56,57].

Osimertinib has already an established value as adjuvant treatment in resected patients with EGFR mutations, based on the ADAURA trial showing a significant improvement in two-year DFS with osimertinib vs. placebo [6,8,62]. When evaluating neoadjuvant targeted therapies, these treatments are regarded as safe and feasible [9,61]. In a subgroup analysis of 68 patients with N2 disease extracted from three studies, mediastinal downgrading was achieved in 14%. No pCR was seen [61]. The EMERGING CTONG 1103 trial did show improvement towards PFS in favour of erlotinib, though recently, no difference in OS was reported [58]. The same conclusion regarding survival was observed in another phase II trial [59].

Another trial showed an improved radical resection rate and a correlation between TP53 mutation burden and PFS. Therefore, performing next generation sequencing is increasingly important in a non-metastatic setting and needs to be further explored [60].

Considering unresectable stage IIIA-N2 disease, more definite recommendations can be provided. Concurrent chemoradiation is advised [19]. In the recent 2021 update of both ESMO and NCCN, durvalumab as consolidation therapy is advised in patients with PD-L1 levels ≥ 1% if no progression is noted [6,8]. This recommendation was based on the PACIFIC trial. Both PFS and OS were significantly higher favouring durvalumab [64]. These same results were seen in the LUN 14-179 trial [65].

Currently, many trials such as the NICOLAS trial [67], the AFT-16 trial [69], the DETERRED trial [70], the EA5181 trial [72], and the PACIFIC2 trial [71] are ongoing, investigating the role of the concomitant use of ICI with concurrent CRT.

## 5. Conclusions

The ideal treatment in case of mediastinal lymph node involvement in stage IIIA-N2 NSCLC is still a source of controversy. Many studies try to elucidate optimal treatment in this heterogenous group. Multidisciplinary teams are of paramount importance to evaluate each case and to determine whether complete resection is feasible. N2 subcategories are yet to be further evaluated and may need to be separately investigated in future trials. Neoadjuvant chemoradiotherapy plus surgery is recommended by recent guidelines in resectable N2 disease; others suggest neoadjuvant or adjuvant chemotherapy. Involvement of multiple N2 stations still remains the most heterogenous group with different recommendations. Immunotherapy and targeted therapy are showing promising results in both response rate and/or DFS. No guidelines can yet be provided, though neoadjuvant immunochemotherapy seems to be superior in stage IIIA-N2 disease. In completely resected N2 disease, PORT is not advised. Adjuvant osimertinib is already recommended in stage IB-IIIa patients with EGFR mutations who underwent a complete resection. In unresectable N2 disease, concurrent chemoradiotherapy still remains the gold standard with durvalumab as consolidation therapy in the case of PD-L1 levels ≥1%.

## Figures and Tables

**Figure 1 cancers-14-01656-f001:**
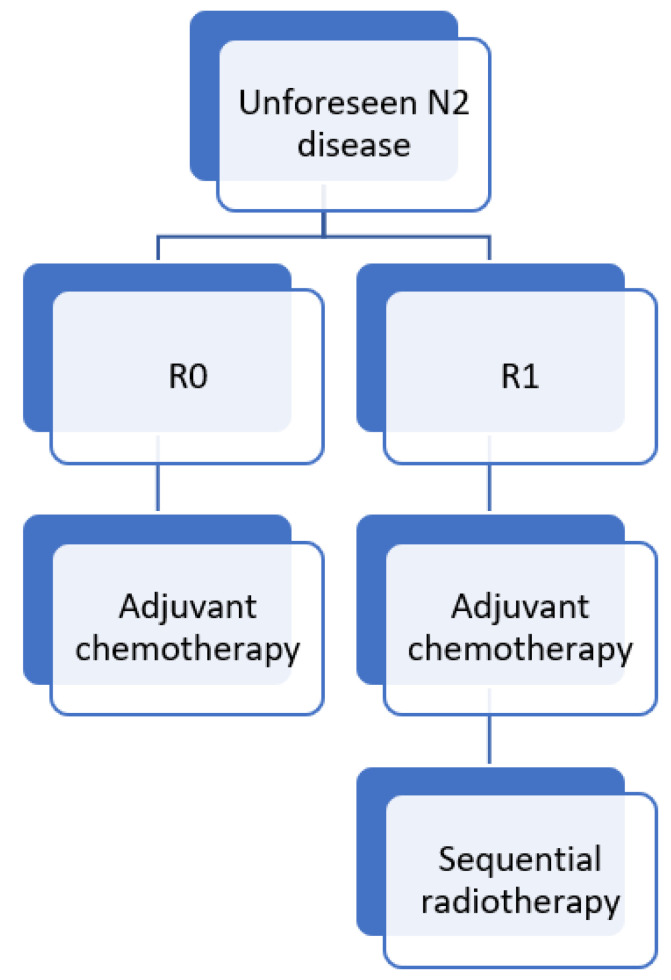
Current treatment algorithm at Antwerp University Hospital for unforeseen N2 disease. R0, complete resection; R1, incomplete resection.

**Figure 2 cancers-14-01656-f002:**
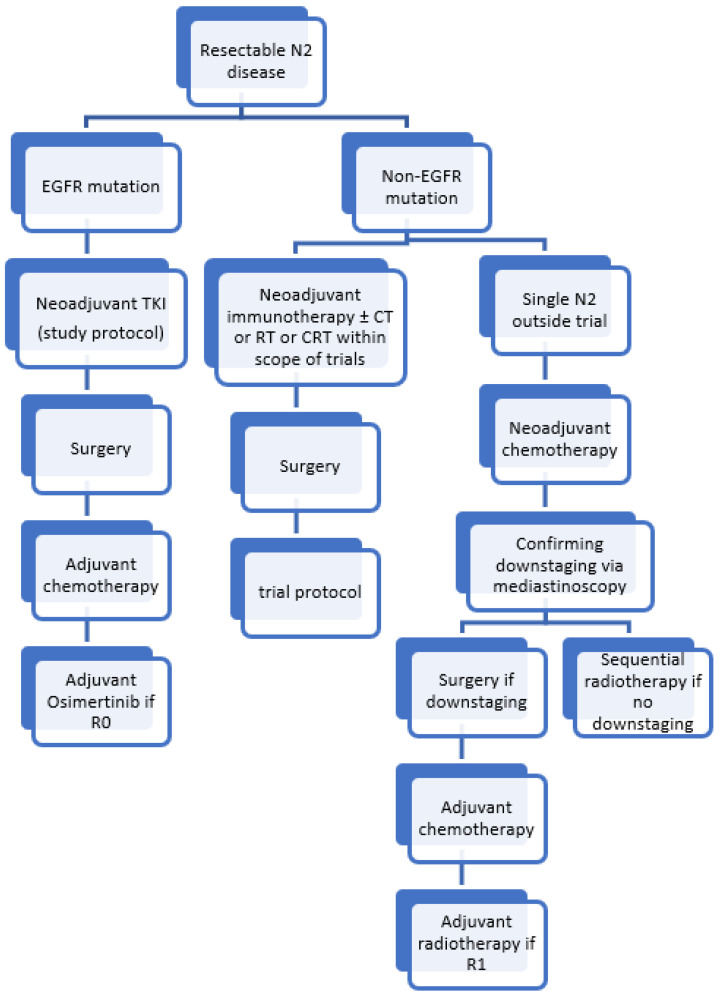
Current treatment algorithm at Antwerp University Hospital for resectable N2 disease. CT, chemotherapy; RT, radiotherapy; CRT, chemoradiotherapy.

**Figure 3 cancers-14-01656-f003:**
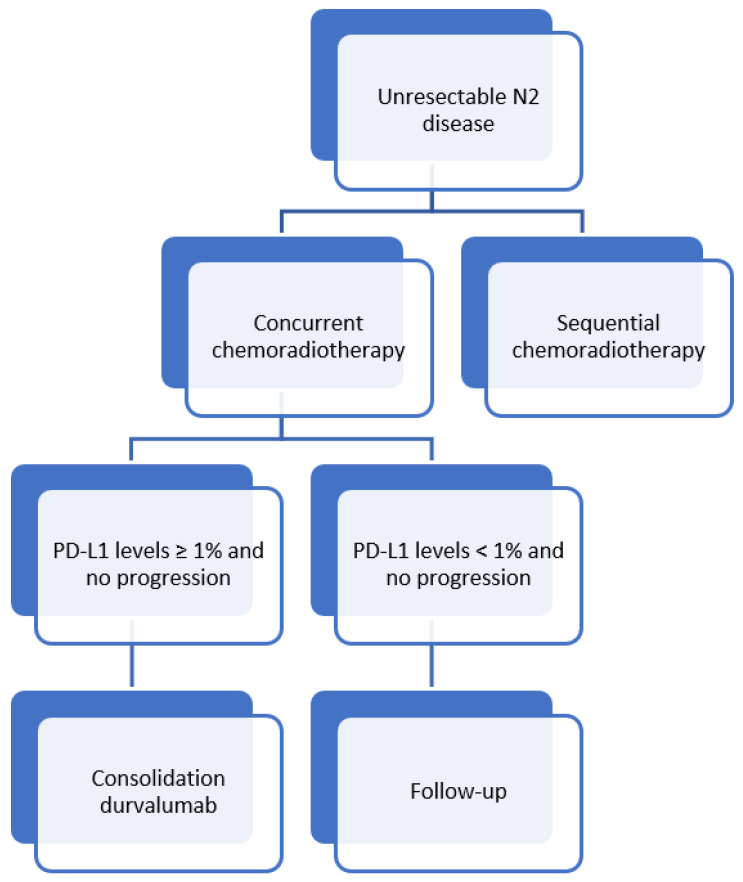
Current treatment algorithm at Antwerp University Hospital for unresectable N2 disease.

**Table 1 cancers-14-01656-t001:** Ongoing trails on neoadjuvant immunotherapy and targeted therapies for resectable NSCLC, including stage IIIA.

Clinical Trial	Phase	Stage	Intervention	(Neo)adjuvant	Estimated Enrollment	Primary Endpoint
Immunotherapy Monotherapy						
NCT04560686	II	I-IIIB	Neoadjuvant Bintrafusp alfa	Neoadjuvant	23	MPR
NCT03197467	II	II-IIIA	Neoadjuvant pembrolizumab	Neoadjuvant	30	pCR, tAEs, radiological response
NCT02818920	II	IB-IIIA	Neoadjuvant and adjuvant pembrolizumab	(neo)adjuvant	35	Surgical feasibility
NCT04062708	II	IIIA-IIIB N2	Neoadjuvant durvalumab + CT, adjuvant durvalumab	neoadjuvant	55	N2 nodal clearance
NCT04379739	II	II-IIIA	Neoadjuvant camrelizumab	neoadjuvant	82	MPR
Combination immunotherapy						
NCT03794544	II	I-IIIA	Neoadjuvant durvalumab vs. durvalumab + oleclumab or monalizumab or danvatirsen	Neoadjuvant	160	MPR
Immunotherapy + CT						
NCT04512430	II	IIIA	Neoadjuvant atezolizumab + bevacizumab + CT, adjuvant atezolizumab	(neo)adjuvant	26	MPR
NCT04326153	II	IIIA	Neoadjuvant and adjuvant sintilumab + CT	(neo)adjuvant	40	DFS
NCT04061590	II	I-IIIA	Neoadjuvant pembrolizumab vs. pembrolizumab + CT	Neoadjuvant	84	Increase tumour-infiltrating cells
NCT03838159	II	IIIA-IIIB	Neoadjuvant Nivolumab + CT vs. CT, adjuvant nivolumab	Neoadjuvant	90	pCR
NCT03800134	III	II-III	Neoadjuvant Durvalumab + CT vs. CT, adjuvant durvalumab	(neo)adjuvant	300	MPR
NCT03456063	III	II-IIIB	Neoadjuvant Atezolizumab + CT vs. placebo + CT	neoadjuvant	374	EFS
NCT04025879	III	II-IIIB	Neoadjuvant Nivolumab + CT vs. placebo + CT, adjuvant nivolumab vs. placebo	(neo)adjuvant	452	EFS
NCT02998528	III	IB-IIIA	Neoadjuvant CT + nivolumab vs. CT vs. nivolumab + ipilimumab	neoadjuvant	350	EFS
Immunotherapy + RT						
NCT03237377	II	IIIA	Neoadjuvant durvalumab + RT vs. durvalumab + tremelimumab + RT	Neoadjuvant	32	Toxicity and feasibility
NCT03217071	II	I-IIIA	Neoadjuvant Pembrolizumab vs. pembrolizumab + RT	Neoadjuvant	40	Change in numbers of infiltrating CD3+ T-cells
NCT04245514	II	III (N2)	Neoadjuvant durvalumab + RT + CT, adjuvant durvalumab + RT	(Neo)adjuvant	90	EFS
Immunotherapy + CRT						
NCT03871153	II	IIIA-N2	Neoadjuvant durvalumab + CRT, adjuvant durvalumab	(neo)adjuvant	25	pCR
NCT02987998	I	IIIA-N2	Neoadjuvant pembrolizumab + CRT, adjuvant pembrolizumab	(neo)adjuvant	9	safety
NCT03694236	II	III	Neoadjuvant durvalumab + CRT	Neoadjuvant	39	pCR
NCT04202809	II	III	Neoadjuvant durvalumab + CRT vs. neoadjuvant CRT, adjuvant durvalumab	(neo)adjuvant	90	PFS
Targeted therapy						
NCT03433469	II	I-IIIA	Neoadjuvant osimertinib	Neoadjuvant	27	MPR
NCT04302025	II	IIA-IIIB	Neoadjuvant and adjuvant alectinib or entrectinib or vemurafenib and cobimetinib or pralsetinib	(Neo)adjuvant	60	MPR
NCT04351555	III	II-IIIB N2	Neoadjuvant osimertinib vs. Osimertinib + CT vs. CT	Neoadjuvant	328	MPR

MPR, major pathological response; pCR, pathological complete response; tAEs, treatment-related adverse events; DFS, disease-free survival; EFS, event-free survival; PFS, progression-free survival.

**Table 2 cancers-14-01656-t002:** Ongoing trails on immunotherapy in unresectable NSCLC, including stage IIIA.

Clinical Trial	Phase	Stage	Intervention	Estimated Enrollment	Primary Endpoint
Immunotherapy + RT					
NCT04013542	I	Unresectable II-III	nivolumab and ipilimumab + RT	20	Safety
NCT03523702	II	Unresectable II-III	Pembrolizumab + RT (if PD-L1 > 50%) vs. CRT (if PD-L1 < 50%)	63	PFS
NCT03644823	II	Unresectable III-IV	Atezolizumab + RT	21	safety
Immunotherapy + CRT					
NCT03631784	II	Unresectable III	Pembrolizumab + CRT	217	Safety, ORR
NCT03102242	II	Unresectable III	Atezolizumab + CRT	64	DCR
NCT04287894	IB	Unresectable II or stage III	Durvalumab + tremelimumab + CRT	34	Safety
NCT03663166	I/II	Unresectable III	Ipilimumab + CRT, consolidation nivolumab	19	Safety, PFS
NCT04026412	III	Unresectable III	nivolumab + CRT, consolidation nivolumab and ipilimumab vs. nivolumab + CRT, consolidation nivolumab vs. CRT, consolidation durvalumab	888	PFS, OS
NCT03285321	II	Unresectable III	CRT, consolidation nivolumab vs. nivolumab and ipilimumab	108	PFS
NCT03693300	II	Unresectable III	CRT, consolidation durvalumab	117	Safety
NCT04380636	III	Unresectable III	CRT + pembrolizumab vs. CRT + pembrolizumab + olaparib vs. CRT + durvalumab	870	PFS, OS
NCT03589547	II	III	CRT, durvalumab, RT, durvalumab	25	Safety, PFS
NCT04085250	II	Unresectable III	CT + nivolumab, CRT, consolidation nivolumab vs. observation	264	PFS
NCT04085250	II	Unresectable III	CRT + atezolizumab vs. consolidation atezolizumab	52	PFS
NCT04092283	III	Unresectable III	CRT + durvalumab vs. consolidation durvalumab	660	OS
NCT03519971	III	Unresectable III	CRT + durvalumab vs. CRT + placebo	328	PFS

PFS, progression-free survival; ORR, overall response rate; DCR, disease control rate; OS, overall survival.

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
