# Peer review of "Multimodality Treatment including Surgery Related to the Type of N2 Involvement in Locally Advanced Non-Small Cell Lung Cancer"

_cancers, 2022, doi:10.3390/cancers14071656_

Round 1
Reviewer 1 Report
This reviewer thanks the authors for all the changes made according to the comments, which qualitatively improve the review.
Reviewer 2 Report
In this manuscript the authors tried to review on multimodality therapeutic options for stage IIIA-N2 NSCLC depending on the extent of nodal involvement.
The Authors performed a “narrative review” evaluating the different situations of mediastinal nodal disease and explaining their one attitude by reporting ad hoc algorithm useful for readers.
Moreover, they summarize the actual attitudes towards the treatment of N2 NSCLC illustrating new protocols and studies evaluating advantages and disadvantages.
They analyzed the role of radiation therapy in this particular setting of patients (N2-NSCLC) reporting evidences on its usefulness.
The Authors should be congratulated for their work; this manuscript is a practical overview on the management on N2-NSCLC and, in my opinion, it is useful for oncologists, radiotherapists and surgeons who deal with NSCLC.
This manuscript is a resubmission of an earlier submission. The following is a list of the peer review reports and author responses from that submission.
Round 1
Reviewer 1 Report
The authors have submitted a review of the latest data on multimodality
treatment options for stage IIIA-N2 locally advanced NSCLC. However, the authors are all from a department of Thoracic and Vascular Surgery and in my opinion such a review requires representatives from the various diciplines involved in multimodal treatment of these patients. Also, the review do not follow Cancers guidelines (PRISMA) for review articles and no information about search strategies are given. Coming from radiotherapy, I get the impression that this review is biased. The interplay between radiotherapy and immunotherapy is for instance not discussed. Also the authors several times present findings as relevant, although not significant (p.2 line 90-92, p.3 line 104-105, p.3 line 141-142, p.5 184-187). I find this suspicius.
Reviewer 2 Report
Multimodality treatment including surgery related to the type of N2 involvement in locally advanced non-small cell lung cancer
- T. Allaeys et al. present a review that aims to summarize different management options for patients presenting mediastinal nodes (N2 disease). While I think this issue is important for surgeons, I am not sure that the topic is of interest to audience of
The review is limited to systematically collecting information from different sources, guidelines, clinical trials, etc. but it lacks critical discussion and directionality.
A major revision is required in order to publish on Cancers of high quality.
- The text becomes difficult to read due to lack of directionality. It is difficult to find the main ideas of each paragraph. The lack of consensus in the treatment of patients is transmitted in the text.
- No new ideas are proposed. Authors just make a statement that a consensus in the guidelines is needed but they do not give their opinion or contribute with anything original.
- As authors are speaking about mediastinal lymph nodes and a specific stage of patients with particular anatomical features, I would suggest some drawing representing this situation to make it easier for the reader to locate the illness. I also suggest the authors to make an effort on graphics to summarize the different therapeutic options.